Translation and psychometric validation of the Persian version of the Questionnaire on Smoking Urges for assessment of craving to smoke among university students

Kazemitabar Maryam maryam.kazemi64@ut.ac.ir ma.kazemi64@gmail.com 1 2
Garcia Danilo Danilo.garcia@icloud.com 3 4 5 6 7
1 Department of Psychology, University of Tehran , Tehran , Iran
2 Promotion of Health and Innovation (PHI) Lab, International Network for Well-Being , Iran
3 Department of Behavioral Sciences and Learning, Linköping University , Linköping , Sweden
4 Centre for Law and Mental Health (CELAM), University of Gothenburg , Gothenburg , Sweden
5 Promotion of Health and Innovation (PHI) Lab, International Network for Well-Being , Sweden
6 Department of Psychology, University of Gothenburg , Gothenburg , Sweden
7 Department of Psychology, Lund University , Lund , Sweden
Patton Bob
Electronic publication date: 2021 Nov 24
Publication date: 2021
Volume: 9
Electronic Location ID: e12531
Received 2021 Mar 29; Accepted 2021 Nov 2
Copyright: ©2021 Kazemitabar et al.
Copyright year: 2021
Copyright holder: Kazemitabar et al.
License: This is an open access article distributed under the terms of the Creative Commons Attribution License, which permits unrestricted use, distribution, reproduction and adaptation in any medium and for any purpose provided that it is properly attributed. For attribution, the original author(s), title, publication source (PeerJ) and either DOI or URL of the article must be cited.
License URL: https://creativecommons.org/licenses/by/4.0/

Keywords: Smoking urges, Smoking cessation, Psychometric properties, Smoking craving, QSU, Questionnaire on smoking urges, Validity and reliability

Funding: The authors received no funding for this work.

==============================
Background

Even though tobacco is one of the most preventable causes of death worldwide, it endangers more than 8 million people yearly. In this context, meta-analyses suggest that a significant part of the general Iranian population over 15 years of age smoke and that there is a need for good screening tools for smoking cravings and urges in Iran. The present study reported the translation and investigated the psychometric properties (i.e., factor structure, validity, and reliability) of the Persian version of the Questionnaire on Smoking Urges (QSU) with 12 items in the Iranian context.

Method

The translation process and content validity of the items were examined entirely in an expert panel using the Content Validity Index. The total sample of participants in which the translated version was tested consisted of 392 (172 female, 220 male, Mage = 22.31 years, SD = 2.90) university students who answered the QSU 12-item at the start of their participation in smoking cessation interventions. The QSU 12-item was firstly translated, then piloted using a subsample of 150 university students and finally validity and reliability of the instrument were investigated using a subsample of 242 participants. We tested the proposed models in the literature, that is, a 1-factor solution and a 2-factor solution with six items on each factor (Factor 1: desire/intention to smoke; Factor 2: relief of negative affect or withdrawal symptoms and anticipation of positive outcome). At last, we tested differences across differences in QSU-scores across different subgroups of individuals based on their demographics.

Results

The results suggested that, in contrast to past studies, a modified 2-factor model, using five items for Factor 1 and 7 items for Factor 2, was the best fitting model (CFI = .95, RMSEA = .09, CI = 90%). Additionally, the QSU 12-item Persian version showed good convergent and divergent validity, internal consistency (Factor 1 = .94, Factor 2 = .97), ICC (average measure ICC = .95, CI = 95%, F(391, 4301) = 20.54, p < .001), concurrent validity (r = .71, p < .01), and discriminant validity (r = −.04, p > .05). Finally, subgroups based on gender, marital status, (un)employment, and educational level did not differed in their responses to the QSU 12-item.

Conclusion

The Persian version of the QSU 12-item has satisfactory psychometric properties and, with a slight modification, it can be considered as a reliable and valid method to estimate smoking urges in the Iranian population. Moreover, the QSU 12-item seems appropriate to measure urge for smoking among groups of individuals with different sociodemographic backgrounds. Importantly, the QSU 12-item differentiates individuals’ desire and intention to smoke from their anticipated relief of negative affect or withdrawal symptoms, which can be important for personalizing interventions targeting individuals who want to quit smoking.

Introduction

Smoking is one of the biggest global health challenges and a high-risk factor for premature death. Even though tobacco is one of the most preventable causes of death worldwide, it is the cause of death of more than 8 million people/year. Above 7 million of those deaths are a direct result from tobacco use and around 1.2 million deaths are non-smokers exposed to second-hand smoke (World Health Organization, 2021). Smoking is also associated with many health issues such as sleep problems (Bellatorre et al., 2017), hypertension, myocardial infraction and respiratory diseases (Gao, Shi & Wang, 2017), type 2 diabetes (Yuan & Larsson, 2019), schizophrenia spectrum disorders (Scott et al., 2018), stroke (Pan et al., 2019), cardiovascular injury (Al Rifai et al., 2017), lower subjective well-being (Churchill & Farrell, 2017), poor mental health (Bang et al., 2017), and etcetera.

One of the most important motives for smoking is craving and urging for it. Craving for smoking is defined as persistent urges, thoughts, or desire to smoke a cigarette (Potvin et al., 2015). It is considered a common symptom of addiction and identified as one of the characteristics of psychoactive substance dependence (American Psychiatric Association, 2013) and is also one of the important causes of maintenance of addiction (Serre et al., 2018) and associated with smoking relapse in individuals who seek smoking cessation interventions (Motschman, Germeroth & Tiffany, 2018; Chatterjee et al., 2016; Killen & Fortmann, 1997). Hereupon, measuring craving is important prior to starting any smoking cessation intervention (Waters et al., 2013; Shiffman, West & Gilbert, 2004) as decrease in craving for smoking would be effective in successful treatments (Enkema & Bowen, 2017).

There are several similar questionnaires measuring the desire and craving for smoking such as Smoking initiation for Women Questionnaire (Shahbazi Sighaldeh et al., 2019), Nicotine Dependence Syndrome Scale (Shiffman, Waters & Hickcox, 2004), Willingness to Quit (Onchonga et al., 2020), and Tobacco Craving Questionnaire (Heishman, Singleton & Moolchan, 2003). Nevertheless, one of the most commonly used measures is the Questionnaire on Smoking Urges (QSU), which was initially developed by Tiffany & Drobes (1991). The QSU comprises 32 items measuring two factors and four categories. Factor 1 indicates (a) the desire to smoke, and (b) anticipation of pleasure from smoking; Factor 2 indicates (c) anticipation of relief from negative affect and withdrawal symptoms and (d) intention to smoke. In the original study, the internal consistency for Factor 1 was estimated to .95 and for Factor 2 to .93 (Tiffany & Drobes, 1991), thus, acknowledging the QSU 32-items as a highly reliable scale. Different shorter versions of the QSU include a 26-item version by Tiffany & Drobes (1991), a 12-item version by Kozlowski et al. (1996), and a 10-item version proposed by Cox, Tiffany & Christen (2001). The QSU 12-item had the highest goodness of fit when compared to both the original 32-item and the 26-item versions (Toll et al., 2004; Dethier et al., 2014). Hence, the 12-item version was not only brief and convenient, but also of great interest for research and clinical practice due to its good psychometric properties.

The QSU 12-item has been adapted, studied, and validated in several languages, including French (Dethier et al., 2014; Guillin et al., 2000), Spanish (Cepeda-Benito & Reig-Ferrer, 2004), Portuguese (Araujo et al., 2007), Chinese (Yu et al., 2010), Italian (Teneggi et al., 2001), and German (Müller et al., 2001). However, it has not yet been validated in Persian. As the matter of fact, to the best of our knowledge, there is no standard and validated instrument for measuring urge for smoking exists in the Persian language. Importantly, meta-analyses implied that a significant part of the general Iranian population over 15 years of age smoke (e.g., Moosazadeh et al., 2013). Thus, suggesting the need for a good screening tool for smoking cravings and urges in Iran. Hence, this study aimed to report the translation and investigate the psychometric properties of the Persian version of the QSU 12-item in an Iranian population. In this endeavor, we analyzed both the 1-factor and 2-factor solutions of the QSU 12-item and investigated its validity (i.e., convergent, divergent, discriminant, and concurrent validity) and reliability (i.e., internal consistency).

Method

Ethical statement

The participants provided informed consent before replying to the questionnaires and sufficient information was provided regarding the aims of the research. They were also assured of the confidentiality of their information. This study received ethical approval from the Department of Psychology, Faculty of Psychology and Educational Sciences, University of Tehran (document reference: 2416-14).

Participants

The data was collected within a period of six months from November 2015 to May 2016, except for several days in March and April during Iran’s New Year holiday. The participants, a convenient sample, consisted of 416 university students with the following criteria: being a cigarette smoker who voluntarily sought to participate in smoking cessation interventions, being a university student, and have started smoking on a daily basis for at least one year. The students who smoked only occasionally were excluded from the study. The participants were students at Tehran University of Art, College of Fine Arts of University of Tehran, Amirkabir University of Technology (Tehran Polytechnic), and Islamic Azad University of Tehran (North Branch).

A total of 24 participants were removed due to incomplete answers (i.e., about 6% attrition). Hence, 392 individuals (Mage = 22.31 years, SD = 2.90) constituted the data that was analyzed in the present study (i.e., a survey response rate of 94%). Regarding gender, 172 participants (43.9%) were females and 220 (56.1%) were males. The average age for females’ first smoking experience was 18.55 years and 18.81 years for males; with a total mean of 18.70 years (SD = 2.62) for both genders. Regarding marriage status, 353 individuals were single and 39 were married. Furthermore, 118 individuals were employed for wages and 274 were unemployed. Finally, a total of 288 individuals (73.5%) stated that they had at least one smoker member in their families besides themselves and 104 individuals (26.5%) stated that they were the only smoker in their family. Concerning educational levels, 246 of the participants were bachelor’s students (62.8%), 133 were master’s students (33.9%), and 13 were associate’s students (3.3%).

Measures

Questionnaire on Smoking Urges 12-item version (QSU 12-item)

The QSU 12-item (Kozlowski et al., 1996) is a self-report that measures urge and craving to smoke with 12 statements that are rated using a seven-point Likert scale ranging from 1 (strongly disagree) to 7 (strongly agree). Four of the 12 items (“Smoking a cigarette would not be pleasant.”, “Even if it were possible, I probably wouldn’t smoke now.”, “I have no desire for a cigarette right now”, and “A cigarette would not taste good right now.”) are scored reversely. In the present study, the QSU 12-item was translated into Persian following the process described in the Procedure section.

Nicotine Dependency Syndrome Scale (NDSS)

The NDSS is a multidimensional scale assessing dependency to nicotine (Shiffman, Waters & Hickcox, 2004). It contains 19 items that are rated using a five-point Likert scale scoring from 1 (completely incorrect) to 5 (completely correct). The NDSS scale has five subscales: drive (craving and withdrawal, and subjective compulsion to smoke), priority (preference for smoking over other reinforces), tolerance (reduced sensitivity to the effects of smoking), continuity (regularity of smoking rate), and stereotypy (invariance of smoking). All items are non-reversed.

Self-Motivation for Smoking Cessation (MSC)

The MSC (Joseph et al., 2005) measures self-motivation for quit smoking. It consists of 12 items with a seven-point Likert scale response ranging from 1 (completely disagree) to 7 (completely agree). This scale comprises two factors with six items each: autonomous regulation and controlled regulation.

Procedure

Translation process of the QSU 12-item

An expert panel consisting of two professors of the English language and two specialists with Ph.D. degrees in psychology translated the original QSU 12-item into Persian and back-translated the items. The translation was conducted for each item with precision and accuracy while trying to keep the meaning of the original items. Then, the items were back-translated into English in order to depict the initial translation by another translator who was not involved in the first translation from English to Persian. Both Persian and English items were compared in detail. Eventually, the first draft of the QSU 12-item Persian version was finalized after some minor revisions and corrections such as grammatical error corrections and providing synonyms for some words that could transfer the meaning of the original items literally and accurately.

This version was administered in a sample of 25 examinees (15 males and 10 females, Mage = 23.6 years, SD = 1.4) to investigate the comprehensiveness and clarity of the items. The questionnaire was also scrutinized in terms of content validity in an expert group including 10 psychology professors. They scored each item regarding clarity and comprehensiveness using the Content Validity Index (Waltz & Bausell, 1981) using a four-point rating scale (1 = bad, to 4 = very good). All 12 items achieved good content validity scores at this stage. The Content Validity Index scores for the items are provided in Table A1.

Data analysis

As the first step, we tested the normal distribution of the data by investigating the skewness and kurtosis. Second, through exploratory factor analysis, the number of factors and items’ factor loadings were checked. We conducted both analyses using SPSS v24. Then, we performed a confirmatory factor analysis using Mplus v8.3 (Muthén & Muthén, 2009). For each analysis, different data subsets were used; a procedure recommended when researchers conduct both exploratory factor analysis and confirmatory factor analysis (Worthington & Whittaker, 2006). Using the maximum likelihood estimation method, we tested three models comprising (1) single-factor solution, (2) two-factor solution, and (3) two-factor solution with six items for each subscale (i.e., original model). The fit indices of the chi-square statistics (Hatcher, 1996), comparative fit index (CFI; Bentler & Wu, 1998), Tucker-Lewis index (TLI; Tucker & Lewis, 1973), standardized root-mean-square residual (SRMR; Jöreskog & Sörbom, 1981), and root-mean-square error of approximation (RMSEA; NE & Cudeck, 1993) were estimated for all three models. Additionally, the multicollinearity between factors was estimated. Sample size for exploratory factor analysis was calculated at least 10 times the items (Nunnally, 1967) and for confirmatory factor analysis it was calculated using GPower v3.1 (Faul et al., 2007). Convergent and divergent validity were also calculated using Fornell & Larcker’s (1981) criterion.

Internal consistency of the subscales and the total construct were measured by Cronbach’s alpha as the measure for reliability (Revelle & Zinbarg, 2009) and intraclass correlation coefficient. Concurrent validity was assessed by comparing the QSU 12-item to the NDSS and estimating the Pearson correlation coefficient between them; the mean scores and standard deviations for both scales were also calculated and reported. For the evaluation of QSU 12-item’ discriminant validity, we estimated the Pearson correlation coefficients between QSU 12-item and the MSC. The differences in responses to the QSU 12-item in subgroups of gender, marriage status, employment status, and educational level were studied as the final analysis.

Results

Normality test of the data

The skewness and kurtosis normality assumptions were tested before conducting the factor analyses. The skewness ranged between −1.07 and −.25 and the kurtosis ranged between −.63 and .71. Hence the distribution of the data was assumed as normal (Hair et al., 2021).

Maximum likelihood exploratory factor analysis

The subsample size of 150 participants was estimated as sufficient (KMO = .93) for conducting the exploratory factor analysis. In addition, the Bartlett’s test of sphericity showed that the items were sufficiently correlated for conducting a factor analysis (chi-square = 2347.32, df = 66, p < .01). The analysis showed that there were two factors with eigenvalues higher than one and that these two factors accounted for 86.47% of the total variance. The first factor determined 61.31% and the second component 25.15% of the total variance (Table 1).

Table 1 Factor loadings of the QSU 12-items using maximum likelihood exploratory factor analysis.

Items	Factors	
	1	2	
QSU1	.95	.38	
QSU2	.92	.36	
QSU4	.94	.42	
QSU7	.87	.29	
QSU9	.93	.39	
QSU10	.89	.31	
QSU12	.94	.36	
QSU3	.34	.92	
QSU5	.36	.85	
QSU6	.33	.87	
QSU8	.35	.93	
QSU11	.34	.89	

The items 1, 2, 4, 7, 9, 10, and 12 were located within one factor and the items 3, 5, 6, 8, and 11 within the other factor (χ2 = 99.90, df = 43, p < .01). The χ2/df = 2.32 indicated a good model fit. All items had factor loadings higher than .85 after rotation converged in three iterations. The extraction method was maximum likelihood and the rotation method was Oblimin with Kaiser Normalization, which is an oblique rotation. Oblique rotation is used when it is assumed that the factors are correlated (Kieffer, 1998); which according to the literature, is the case for the QSU 12-item’ factors. Next, a confirmatory factor analysis was performed in order to acquire adequate support for the results from the exploratory factor analysis.

Confirmatory factor analysis

The confirmatory factor analysis was performed for the single-factor solution, for the proposed modified two-factor solution (based on the exploratory factor analysis), and for the original two-factor solution using maximum likelihood estimation method in Mplus v8.3 on a sample of 242 participants. Thus, as suggested in the literature (Worthington & Whittaker, 2006), the subsamples used in the exploratory factor analysis and the confirmatory factor analysis were not the same. The comparison between these three models showed that the modified two-factor model had a better fit. In this model, unlike the original model proposed by Tiffany & Drobes (1991), item number 9 (I have an urge for a cigarette) fitted better on Factor 2 instead of Factor 1 by showing a higher factor loading for this item on Factor 2 (.91) rather than Factor 1 (.21). The modified two-factor model exhibited better fit indices compared to the single-factor model and the original two-factor model (see Table 2).

Table 2 Fit indices for the confirmatory factor analysis of the 1 and 2 factor models of the QSU 12-item.

Model	χ 2	df	χ2/df	ρ	RMSEA (CI = 90%)	CFI	TLI	SRMR	
1-factor	921.81	54	17.07	.00	.25	.76	.70	.14	
2-factor (modified)	221.88	53	4.18	.00	.09	.95	.94	.04	
2-factor (original)	562.17	53	10.60	.00	.19	.86	.82	.12	
Notes.

RMSEA, Root Mean Square Error of Approximation; CFI, Comparative Fit Index; TLI, Tucker-Lewis Index; SRMR, Standardized Root Mean Square Residual.

All three models had significant chi-square statistics (p < .01), but the modified two-factor model had better chi-square goodness of fit value. The RMSEA for the single-factor model (.25) and the original two-factor (.19) were, however, unacceptable. In the modified two-factor model, the RMSEA (CI = 90%) was marginal (.09). The CFI values for single-factor model (.76), modified two-factor model (.97), and original two-factor model (.86) were calculated. The general acceptable CFI value should be above .95 which is achieved by the modified two-factor model in our CFA analysis. The SRMR value in the single-factor (.14) and the original two-factor (.12) models were high, which suggested a poor fit for the data, while in the modified two-factor model it was very low (.03), thus, indicating a low residual error in this specific model. According to Hu & Bentler (1999), a CFI and TLI ≥ .95, a SRMR ≤ .08, and a RMSEA ≤ .06 are needed to conclude that the hypothesized model has a relatively good fit to the observed data. Therefore, we argue that the modified two-factor model had the best fit rather than the single-factor and the original two-factor models. The multicollinearity between Factor 1 and Factor 2 in the modified two-factor model was estimated as low (Tolerance = 1.00, VIF = 1.00). Figure 1 represents the modified two-factor model of the QSU 12-item.1 Factor loadings, standard errors, t-values, and p-values of the items are presented in Table A2. The next analyses were conducted using the modified two-factor solution.

Figure 1 Path diagram for the Persian version of the QSU 12-item.

The values out of parentheses indicate factor loadings, the values inside the parentheses indicate standard error of estimates, and the numbers in rectangles indicate items’ number.

Convergent and divergent validity

Table 3 shows composite reliability (CR), average variance extracted (AVE), and AVE squared for Factor 1 and 2. According to Fornell & Larcker (1981), convergent validity is approved if AVE > .5 and AVE < CR and divergent validity is confirmed if the AVE square root of each factors is greater than the correlations between factors (r = .51). The obtained results suggested that the QSU 12-item’ factors had sufficient convergent and divergent validity.

Table 3 CR, AVE, and AVE2 of the Factors of the Persian version of the QSU 12-item.

Factors	CR	AVE	AVE2	
1	.95	.79	.62	
2	.98	.87	.75	
Notes.

CR, composite reliability; AVE, average variance extracted.

Reliability

Internal consistency, as a measure of reliability, was assessed using Cronbach’s alpha (Bonett & Wright, 2015) for each factor independently as recommended elsewhere (e.g., Kazemitabar et al., 2020). The Cronbach’s alphas for Factor 1 (α = .94) and Factor 2 (α = .97), and for the total construct (α = .95) suggested high internal consistency between the items in each factor. The intraclass correlation coefficient was also measured using a two-way random effect model (average measure ICC = .95, CI = 95%, F(391,4301) = 20.54, p < .001) and indicated that the QSU 12-item had high reliability. Table 4 shows that the items within each factor had high correlations (r ≥ .71) with each other and lower correlations (r ≤ .53) with items from the other factor. Moreover, the correlation between the two factors (r = .51, p < .001) showed there was a moderate positive relationship between them.

Table 4 Correlation matrix for the Persian version of the QSU 12-item (modified 2-factor solution model).

	Factor 1	Factor 2	
Item	3	5	6	8	11	1	2	4	7	9	10	12	
3	–												
5	.71	–											
6	.75	.76	–										
8	.80	.71	.73	–									
11	.78	.72	.83	.78	–								
1						–							
2						.89	–						
4						.89	.85	–					
7						.81	.82	.84	–				
9						.89	.86	.89	.80	–			
10						.83	.82	.81	.83	.82	–		
12						.88	.87	.88	.82	.87	.84	–	

Concurrent validity: QSU 12-item and NDSS

The NDSS measurement properties has been investigated in several studies. For example, Costello et al. (2007) studied the factor analysis of this scale in an American college sample. Their results showed a CFI equal to .95 and an RMSEA equal to .06. Sterling and colleagues (2009) also measured the psychometric properties of the NDSS among teens and the results indicated alpha ranging from .64 to .92, CFI = .94, and RMSEA = .09; hence, suggesting a good fit to their data. In other words, it had high validity and reliability to measure the nicotine dependency.

In the present study, the Pearson correlation between the QSU 12-item and the NDSS was relatively high and positive (r = .71, p < .01), that is, the higher the smoking urges and cravings are, the higher the nicotine dependency is. That is, indicating a fair concurrent validity between these two scales. The mean score for the NDSS was 65.00 (SD = 9.23) and for the QSU 12-item the mean score was 64.99 (SD = 12.95).

Discriminant validity: QSU 12-item and MSC

Previous studies showed that the MSC is valid and reliable. For instance, Çelik (2014) measured the factorial structure of the MSC in a Turkish sample, the results showed a good fit to the data (CFI = .93, GFI = .92, RMSEA = .08, SRMR = .06) with good internal consistency (.81 and .76). In the present study, the correlation between the MSC and NDSS was nonsignificant (r = −.08, p > .05). Likewise, the correlation between the QSU 12-item and the MSC was also nonsignificant (r = −.04, p > .05). In other words, there was no relationship between them and despite being slightly negative, which was expected, the correlation between the MSC and the QSU 12-item was extremely low.

Differences between groups: gender, marital status, education, and employment

The normality of residual assumption for conducting parametric tests showed non-normality of the residuals for the different demographic groups. Therefore, non-parametric analyses, Mann–Whitney and Kruskal–Wallis, were used to investigate differences in craving and urge for smoking between individuals with different gender, marriage status, educational level, and employment status. No significant differences were detected within subgroups for the total QSU 12-item score (p > .05) or for Factor 1 (i.e., desire/intention to smoke) and Factor 2 (i.e., relief of negative affect or withdrawal symptoms and anticipation of a positive outcome) (p > .05) (see Table 5).

Table 5 Mann–Whitney and Kruskal–Wallis tests for comparing subgroups in smoking urges and craving as measure by the QSU 12-item (modified 2-factor solution model).

QSU	Groups	N	Mean rank	Mann–Whitney test	
					U	p-value	
	Gender	Female	172	194.35	18550.00	.73	
		Male	220	198.18			
	Marriage status	Single	353	197.08	6679.00	.76	
		Married	39	191.26			
	Employment status	Employed	118	191.17	15537.50	.54	
		Unemployed	274	198.79			
	Educational level		N	Mean rank	Kruskal–Wallis (Chi-Square)	p-value	
		Associate	13	212.23	.51	.77	
		Bachelor	246	198.12			
		Master	133	191.97			
Factor 1	Gender	Female	172	199.73	18364.00	.61	
		Male	220	193.97			
	Marriage status	Single	353	197.66	6473.00	.54	
		Married	39	185.97			
	Employment status	Employed	118	203.63	15324.50	.41	
		Unemployed	274	193.43			
	Educational level		N	Mean rank	Kruskal–Wallis (Chi-Square)	p-value	
		Associate	13	193.77	.05	.97	
		Bachelor	246	197.47			
		Master	133	194.97			
Factor 2	Gender	Female	172	193.71	18440.000	.666	
		Male	220	198.68			
	Marriage status	Single	353	197.52	6522.500	.590	
		Married	39	187.24			
	Employment status	Employed	118	189.13	15296.500	.397	
		Unemployed	274	199.67			
	Educational level		N	Mean rank	Kruskal–Wallis (Chi-Square)	p-value	
		Associate	13	229.92	1.758	.415	
		Bachelor	246	198.62			
		Master	133	189.31			

Discussion

The present study reported the translation and examined the psychometric properties of the Persian version of the QSU 12-item in an Iranian population of university students. This was, as far as we know, the first translation and validation study of the QSU 12-item in the Persian language. For this purpose, the translation process and content validity were examined entirely by a panel of expert panel, who judged the translation as satisfactory. Then, the exploratory factor analysis found two factors with five items that loaded on one of the factors and the other seven items on the second factor (loadings ranging from .85 to .95)—that is, a modified two-factor solution. On this basis and past findings from the literature, we tested three models (i.e., single-factor, original two-factor, and the proposed modified two-factor solutions) through confirmatory factor analysis. The goodness of fit indices obtained from each model suggested that the modified two-factor structure had the best fit. Moreover, Cronbach’s alpha coefficients showed high reliability for the total score and for the subfactors in this modified two-factor solution. Our findings are partially consistent with past studies that have confirmed a two-factor solution for the QSU 12-item (e.g., Tiffany & Drobes, 1991; Toll et al., 2004; Dethier et al., 2014; Cepeda-Benito & Reig-Ferrer, 2004).

Nevertheless, the exploratory and confirmatory factor analyses indicated that Factor 1 (desire/intention to smoke) consisted of five items and that Factor 2 (relief of negative affect or withdrawal symptoms and anticipation of positive outcome) consisted of seven items. This was slightly different from past studies suggesting six items for each factor (e.g., Kozlowski et al., 1996; Toll et al., 2004). More specifically, item 9 “I have an urge for a cigarette” in the original version belonged to Factor 1 (desire/intention to smoke), but in the modified two-factor solution in the present study, item 9 had significantly higher factor loading on Factor 2 (relief of negative affect/withdrawal symptoms). This finding was, however, in line with the validation of the QSU 12-item French version (Dethier et al., 2014), in which this very same item also loaded on the Factor 2 (relief of negative affect/withdrawal symptoms). According to these researchers, this was due to a translation error by the original French translators (i.e., Guillin et al., 2000) when they developed the French QSU—the word “urge” was translated to suggest an “urgent need” rather than a “strong desire” (Dethier et al., 2014). Thus, making the item semantic content more closely related the concept measured in Factor 2, the anticipation of relief of negative affect and withdrawal symptoms. Although, this might be the case for the Persian version developed here, our translation (see Table A3) is more attuned with “having a strong desire”, thus, we attribute the modification of the two-factor solution to the actual psychometric properties of the QSU 12-item.

In this context, the modified two-factor model showed high goodness of fit (CFI = .95, TLI = .94) while the error value was marginal (RMSEA = .09, CI = 90%) (cf. Hair et al., 2010). Browne & Cudeck (1992) suggested that values of .08 or lower for RMSEA demonstrate a reasonable fit. Some other researchers, however, have more strict criteria for RMSEA cutoff points and propose the value of .05 or less as acceptable (Kyle, 1999); while there is also a moderate cutoff of .06 provided by Hu & Bentler (1999). Some other studies have shown that increasing the number of variables resulted in a decrease in the RMSEA value (Bentler & Wu, 1998; Breivik & Olsson, 2001). Importantly, Kenny & McCoach (2003) described that the decline in RMSEA was a result of decreasing the ratio of chi-square to its degree of freedom. Thus, the small number of observed variables (i.e., items) in the QSU 12-item might explain the relatively high value for RMSEA in the present study for the modified two-factor solution that we recommend. Depending on the cut off used for RMSEA, however, this might be a limitation with our proposed model.

Moreover, there are arguments concerning the use of different approaches to exploratory and confirmatory factor analyses in both scale construction studies and validation studies. For example, while some recommend exploratory factor analysis as an intermediate stage before conducting a confirmatory factor analysis (Brown, 2015), other researchers suggested employing either one or the other, but definitely not both type of analysis (Kline, 2015). However, using exploratory factor analysis followed by confirmatory factor analysis is one of the most common methods applied for tool development and validation studies (Worthington & Whittaker, 2006). Many validation studies recommend to, as performed in the current study, first conduct an exploratory factor analysis to measure the underlying factor structure, followed by a confirmatory factor analysis using different samples (Costello & Osborne, 2005; Henson & Roberts, 2006; Worthington & Whittaker, 2006).

Furthermore, the maximum likelihood exploratory factor analysis, used here, and principal factor analysis are two approaches that have been regularly used in exploratory factor analysis. Although, there is another recent method named “regularized exploratory factor analysis” suitable for small samples, in which only one parameter is supposed to be estimated (Jung & Takane, 2008). Principal component analysis is wrongly considered as factor analysis as it does not distinctly assess measurement error. In contrast to principal factor analysis, the maximum likelihood exploratory factor analysis we used here explicitly considers the measurement errors of the observed variables. Also in this line, principal factor analysis does not differentiate the variances and explain all variance in the model while exploratory factor analysis explains common variance. Finally, exploratory factor analysis is not based on a testable model, hence, it is not able to accept or reject a model fit (Norris & Lecavalier, 2010). For all these reasons, we found maximum likelihood exploratory factor analysis as the best suitable approach in our study.

We investigated the QSU 12-item’ construct validity using concurrent and discriminant validity. We found that the direction and magnitude of the correlation between comparable instruments showed that the Persian QSU 12-item’ modified two-factor solution was a valid scale to measure urge for smoking among university students—craving and urges for smoking as measured by the Persian QSU 12-item was associated with nicotine dependence and as nicotine dependence, it was weakly and negatively associated with intention to quit smoking. Last but not the least, how a scale performs among subgroups of a sample is important as it reveals whether it measures the construct among subgroups sufficiently. In this regard, subgroups of females and males, married or single, employed or unemployed, and individuals with different educational levels did not differed in their responses to the QSU 12-item. Thus, the questionnaire seems appropriate to measure urge for smoking among these subgroups.

Limitations, future directions, and implications

Future validation studies of the Persian QSU 12-item should measure test-retest reliability, which is suggested to be a stronger measure to estimate reliability than just, as we did here, reporting Cronbach’s alpha (Taber, 2018). Measurement of beta (Revelle & Zinbarg, 2009) or omega (Al Nima et al., 2020) as measures of reliability are also recommended. Since the population in this study was limited to university students, we suggest researchers to include more diverse populations in future studies. In addition, measuring cross-cultural validation and measurement invariance for language or ethnicity would be helpful. Perhaps this might help to explain the issues with item 9, which loaded in Factor 2 instead of Factor 1 as in the original English validation.

Nevertheless, despite these limitations, the QSU 12-item Persian version developed in this study can be utilized in various sectors, such as, health-related clinics, organizations encouraging employer to quit smoking, schools, and research institutions with the aim of screening and diagnosing for smoking craving and urges of individuals seeking treatment. This questionnaire can also be employed in intervention and prevention studies using experimental or quasi-experimental designs.

Conclusion

Our findings suggested that the Persian version of the QSU 12-item has satisfactory psychometric properties and with a slight modification, it can be considered as a reliable and valid method to estimate smoking urges in the Iranian population. Moreover, the QSU 12-item was useful to predict nicotine dependence and seems appropriate to measure urge for smoking among groups of individuals with different sociodemographic backgrounds. Importantly, the QSU 12-item differentiates individuals’ desire and intention to smoke from their anticipated relief of negative affect or withdrawal symptoms, which can be important for personalizing interventions targeting individuals who want to quit smoking. We argue that this is important, especially in light of the current and future challenges of the 21st century. For instance, during the current pandemic, individuals diagnosed with substance use disorders have had a higher risk of COVID-19 and several serious diseases (Wang et al., 2021). In such conditions, is not only our health at stake, but it is difficult to make healthy self-directed choices necessary to be resilient during these challenges (Wong & Cloninger, 2010). Thus, we need to identify methods to measure and understand what makes people crave for unhealthy ways of handling stress, anxiety, fear, depression, or plain boredom (Cloninger, 2004; Cloninger, 2013).

Supplemental Information

Supplemental Information 1 QSU - raw dataset

Click here for additional data file.

Appendix 1

10.7717/peerj.12531/table-A1 Table A1 The expert panel judgement on content validity of the Persian version of the QSU 12-item.

Criteria	Items	Range	CVI	
Relevance	1	2.0–4.0	.8	
2	3.0–4.0	1.0	
3	3.0–4.0	1.0	
4	2.0–4.0	.9	
5	3.0–4.0	1.0	
6	3.0–4.0	1.0	
7	3.0–4.0	1.0	
8	3.0–4.0	1.0	
9	1.0–4.0	0.8	
10	2.0–4.0	.9	
11	3.0–4.0	1.0	
12	2.0–4.0	.8	
Comprehensiveness	1	3.0–4.0	1.0	
2	3.0–4.0	1.0	
3	2.0–4.0	.9	
4	3.0–4.0	1.0	
5	3.0–4.0	1.0	
6	3.0–4.0	1.0	
7	2.0–4.0	.9	
8	2.0–4.0	.9	
9	3.0–4.0	1.0	
10	1.0–4.0	.8	
11	2.0–4.0	.9	
12	1.0–4.0	.8	
Clarity	1	3.0–4.0	1.0	
2	3.0–4.0	1.0	
3	2.0–4.0	.9	
4	3.0–4.0	1.0	
5	2.0–4.0	.9	
6	3.0–4.0	1.0	
7	2.0–4.0	.8	
8	1.0–4.0	.8	
9	3.0–4.0	1.0	
10	2.0–4.0	.9	
11	3.0–4.0	1.0	
12	1.0–4.0	.8	
Notes.

A Likert four-point scale was used (1 = bad, 2 = rather bad, revision needed, 3 = good but minor revision is needed, and 4 = very good); CVI = Content Validity Index.

Appendix 2

10.7717/peerj.12531/table-A2 Table A2 Persian version of the questionnaire on smoking urges 12-item.

Factor	Items	F. L.	S. E.	t-value	p-value	
F2	1. Nothing would be better than smoking a cigarette right now.	.946	.008	122.016	.000	
F2	2. Smoking would make me less depressed.	.928	.010	94.713	.000	
F1	3. Smoking a cigarette would not be pleasant.R	.837	.022	38.519	.000	
F2	4. All I want right now is a cigarette.	.938	.009	107.739	.000	
F1	5. Even if it were possible, I probably wouldn’t smoke now. R	.804	.025	32.054	.000	
F1	6. I have no desire for a cigarette right now. R	.903	.015	61.068	.000	
F2	7. Smoking now would make things seem just perfect.	.886	.015	60.440	.000	
F1	8. A cigarette would not taste good right now. R	.834	.022	37.657	.000	
F2	9. I have an urge for a cigarette.	.938	.008	114.791	.000	
F2	10. I could control things better right now if I could smoke.	.888	.009	108.509	.000	
F1	11. I am going to smoke as soon as possible.	.925	.013	73.548	.000	
F2	12. I would do almost anything for a cigarette now.	.931	.009	98.755	.000	
Notes.

R, Reversed items; F. L., Factor Loading; S. E., Standard Error.

Appendix 3

10.7717/peerj.12531/table-A3 Table A3 Persian version of the questionnaire on smoking urges 12-item in Persian (Farsi).

	

Additional Information and Declarations

Competing Interests

Author Contributions

Human Ethics

Data Availability

1 The items 3 (Smoking a cigarette would not be pleasant.), 5 (Even if it were possible, I probably wouldn’t smoke now.), 6 (I have no desire for a cigarette right now.), 8 (A cigarette would not taste good right now.), and 11 (I am going to smoke as soon as possible.) were placed in Factor 1. The items 1 (Nothing would be better than smoking a cigarette right now.), 2 (Smoking would make me less depressed.), 4 (All I want right now is a cigarette.), 7 (Smoking now would make things seem just perfect.), 9 (I have an urge for a cigarette.), 10 (I could control things better right now if I could smoke.), and 12 (I would do almost anything for a cigarette now.) were located in Factor 2. The Persian version of the QSU-12 item (in Farsi) is presented in Table A3.

The authors declare there are no competing interests.

Maryam Kazemitabar conceived and designed the experiments, performed the experiments, analyzed the data, prepared figures and/or tables, authored or reviewed drafts of the paper, and approved the final draft.

Danilo Garcia analyzed the data, prepared figures and/or tables, authored or reviewed drafts of the paper, and approved the final draft.

The following information was supplied relating to ethical approvals (i.e., approving body and any reference numbers):

This study received ethical approval from the Department of Psychology, Faculty of Psychology and Educational Sciences, University of Tehran (Document reference: #2416-14).

The following information was supplied regarding data availability:

The raw data are available in the Supplemental File.

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
