# Peer review of "Translation and psychometric validation of the Persian version of the Questionnaire on Smoking Urges for assessment of craving to smoke among university students"

_PeerJ, doi:10.7717/peerj.12531_

## Round 0.1 · original submission · Major Revisions

Thank you for your submission. Please address the comments and suggestions provided by our three reviewers. There is some work to be done before the paper is suitable for publication.

Reviewer 1 ·

Basic reporting

Literature Review can be improved.
Line 89-91: Author stated that QSU has been translated in different languages since this is a validation article why the validity (brief information) in different languages were not provided e.g., French (Dethier et al., 2014; Guillin et al., 2000), Spanish (Cepeda-Benito & Reig-Ferrer 2004), Portuguese (Araujo et al., 2007), Chinese (Yu et al., 2010), Italian (Teneggi et al., 2001), and German (Müller et al., 2001).

Line 93: Strengthen the rationale of this study is to encourage. i.e., line 93 Thus, suggesting the need for both suitable guidelines and also good screening for smoking cravings… smoking in Iran. This manuscript does not deal with guidelines but more towards validating QSU questionnaire. The word suitable guidelines should be omitted.

Line 106: “participation rate”, change to survey response rate.

Line 126: …urge, and .82 for expectancy .82- this is redundant

Line 125-126, 184: There is inconsistency throughout this manuscript in terms of numerical system (for example: either author use .71-US system or 0.71 alternative numerical system). Choose one system.

Experimental design

Line 156: Content Validity Index (Waltz & Bausell, 1981), can you include the proof of CVI information in your attachment?

RESULTS
-Add EFA results table especially factor loadings
-Add information of χ²/df in Table 1.
-Figure 1, Factor 1, and Factor 2 must be named.

Line 228-229 The factor loadings, standard errors, t-values, and p-values of the items are presented in Appendix. I could not find the information in Appendix

Reliability
Line 216-218, 223 Why are these words (RMSEA, CFI, TLI, SRMR) in italic?

Line 242: Why is … Cronbach’s alpha (Bonett & Wright, 2015) in italic?

Line 245-246 "For Factor 1 it was 0.94, for Factor 2 it was 0.97, and for the total construct Cronbach’s alpha was 0.95." I have some concerns as the reliability of Factor 2 is extremely high.
Please read Tavakol and Dennick (2011)
If alpha is too high it may suggest that some items are redundant as they are testing the same question but in a different guise. A maximum alpha value of 0.90 has been recommended (Streiner, 2003).

Validity of the findings

Impact and novelty are well-established but there are a few issues that need to be addressed.

DISCUSSION
Line 307-312-briefly discussed Factor 1 (5 items) and Factor 2 (7 items) but did not provide an explanation of the rationale as to why suddenly 1 item from Factor 1 loaded on Factor 2. An explanation or rationale is needed to justify this issue.

Line 322: Hu and Bentler (Hu & Bentler, 1999) changed to Hu and Bentler (1999).

Line 324: Kenny and McCoach (Kenny & McCoach, 2003) changed to Kenny and McCoach (2003)

Line 332: confirmatory factor analysis (eg., Brown, 2015), some researchers (e.g., Kline, 2015) Delete e.g.,

Line 361: might also need to use Beta (e.g., Revelle & Zinbarg, 2009) or Omega (e.g., Al Nima, 2020)… Delete e.g.,

Line 338-339: Worthington & Whittaker, 2006; Costello & Osborne, 2005; Henson & Roberts, 2006) changed to (Costello & Osborne, 2005; Henson & Roberts, 2006; Worthington & Whittaker, 2006).

Additional comments

Reference needs some attention. In-text citation needs attention.
Proofreading is stringly encourage.
There is major inconsistency throughout this manuscript in terms of the numerical system (for example: either use .71 (US system) or 0.71 alternative numerical system)

Reviewer 2 ·

Basic reporting

Dear authors:
I have now reviewed your paper and recognize your manuscript addresses the interesting research question (psychometric validation). However, I can suggest several weaknesses in the manuscript.
* The quality of presentation.
* The presentation of manuscript contains unnecessary description and self-assertion (not supported by the current knowledge, not properly cited)
* Short paragraphs and lack of consistency between paragraph topics
Abstract
1. Abstract is not balance.
2. Abstract is not informative. For example, number of male and females, Cronbach coefficient, goodness of fit information can be added to abstract.

Introduction
3. The first paragraph, Line 51, 52, the statements must be cited. Also, when the authors talk about the Western and Eastern communities, such classifications must be relevant for study. However, we do not any specific design and outcomes related western or eastern. So, i suggest to delete some phrases.

4. Authors must describe to justify the need for Persian version of measure. For example, if measures of the construct exist in the literature, explain the value added by your new scale. Revision requires theoretical understanding what is essential in the instrument for the targeted population. How might the new measure enhance the substantive knowledge base?

5. In line 56, the authors must cited for both causes and associated, separately, or use one of the verbs.
6. The phrase of “in this context” is vague. Please, speak clearly. In which context?
Also, the measure, dimensions, and characteristics must be explained.

The final paragraph of introduction must be to aim and objectives.

Experimental design

Method
7. The sampling must be more explained.
8. The study period must be added
9. The authors must clearly stated inclusion and exclusion criteria.
10. Mean values must be stated with unit. Age mean values unit is age.

11. Did you conduct a pilot or preliminary study before the main validation study to refer what is important a scale modification?
* If your answer is positive, please describe how you conduct.

12. Please describe more clearly about the statistical analysis. What is your criteria about the significant results? What is your plan to reach the aims?

13. Your translation process did not meet standard guidelines. For example, Translation back-translation process, how did you conduct the translation back-translation process?
14. Please, Report expectation related to the reliability of measures,
15. How you judged the normality?
16. As you aware, EFA and CFA methods do not suggested simultaneously. However, you did.

17. Report the specific sampling strategy taken
18. Please report multicollinearity test
19. For all measure, report internal consistency, and higher scores demonstrations.

Validity of the findings

Results
20. Do not use of “We”. It is not good idea.

21. Please use the APA or Journal Guideline in the report.

22. Results of your study were not well-organized. I recommend

* To concise the result, the correlation results can be illustrated in the table.
* Describe your strategy and fit indices Criteria,
* RMSEA must be accompanied by the 90% CI
* In the results section, clearly state the results according to sequence in statistical analyses

23. Report sufficient descriptive statistics—¬including means, standard deviations,
24. SD has not unit (e.g., years). Please revise.
25. Please provide more information about the internal consistency the overall measure, and its dimensions, separately. (Cr or omega)
26. Please describe how you assessed discriminate validity.
27. Test-Retest Reliability is important to present evidence for new scale.
28. Was the sample size included in the analysis adequate? For continuous scores:
29. Was an intraclass correlation coefficient (ICC) calculated?
30. Divergent validity, convergent validity, and AVE values must be generated. Currently, i cannot make a decision about the validity.
34. Please, conduct a multicollinearity test.
35. Please, address to categorical variable. (e.g., Gender, in sample). For example, is there statistically significance difference between male and females?
37. Add the new Persian version of the scale in appendix.
38. Report the concept(s) you are measuring. The exception to this rule is if you are specifically studying the properties of the scale in question rather than the concept(s) it measures.
39. Put a leading zero for indices that can take values greater than plus 1 or lower than minus 1 such as means, standard deviations, b, beta, and standard error. Do not put a leading zero for indices such as r, alpha, and p. However, journal rules may vary from this but they will then edit and change to their own liking.
40. Report two decimal places in general but three for p values and one for percentage values. At minimum, make sure to be consistent within different indices.
41. Please add the legend for figures.
Discussion
42. Evaluate the strengths and limitations of the work.
43. Conclusions are inadequately clear or sound or supported.



Reference
44. Your references are included several Persian refs, that they cannot read by non-Persian speakers. Please, use the core references instead of those.

Additional comments

DEAR authors
i deeply suggest that you report the study based on STROBE checklist.
also; while your scale is related to health setting,we must be more sensitive about the validation.

Reviewer 3 ·

Basic reporting

The title seems incomplete without the target group and the aim to use the questionnaire. It is suggested to change the title into Translation and Validation of the Persian Version of the Questionnaire on Smoking Urges for Assessment of craving to smoke among university students.

The content of the abstract is inadequate to reflect the robustness of the method and synchronization of the findings. Do you do pre-testing analysis before proceed to confirmatory factor analysis? It is important steps in a translated version of the questionnaire, to ensure the content was well understood by the readers. You need to explain briefly the nature of the questionnaires that you had translated ie the number of items, the scoring or rating system applied, and how the confirmatory factor analysis was performed. Why do the respondents need to answer the Nicotine Dependency Syndrome Scale and the Self-Motivation for Smoke Cessation as well? Does it related to the topic of the study? Your statement in the result section needs to be briefed and consolidated. Your current findings are confusing.

The focus of the study was not well explained. Why the questionnaire is needed? Why can’t you use the available tool like Fagerstrom test and others that mentioned in the text? What is the strength of the chosen questionnaire as compared to others? What kind of theory supports your justification to use the questionnaire?

Experimental design

Study setting and participants: What is your study design? How did you determine the sample size for the study? How did you do the sampling method? We usually do not describe the findings in the method section. Thus, please delete the statement line 104 – 115. How long the data collection took place?

Validation measures: You need to explain in detail the translation and validation process which include 10 stages starting from preparation, forward translation, reconciliation, backward translation, backward translation review, harmonization, cognitive debriefing, review of cognitive debriefing result, and finalization, proof-read and finally preparing the final report. The information for NDSS and MSC might not be related to this study yet.

Data analysis: Did you check for multicollinearity between factors?

Any evidence of ethical clearance from the institution? Please provide the number of approval of ethical clearance.

Validity of the findings

The sociodemographic data is important, please include it in your result section (bring all the information from the method section). Please kindly explained the importance of Table 3. Why the plausible differences between individuals important in validation study?

---

## Round 0.2 · Major Revisions

Thank you for the revised paper. One reviewer (and I agree) notes that there are some further improvements to be made, in particular with the organisation of the manuscript, and ensuring that the discussion encompases all observed results and that the limitations of your work are further explored.

Reviewer 2 ·

Basic reporting

Dear authors
I found the structure of the manuscript difficult the follow, somehow disorganized and scattered. Therefore, I suggest that the author(s) be very precise and simple on what they wanted to say/what they aimed. The basic structure of comparative research should be followed while providing the literature and purposes. The authors can apply the logical flow sample below to all manuscripts.

Also, statistical results such as fit indices in introduction section is bad idea.

Experimental design

Please, state the standard method that you utilized for transcultural process.
You must follow the standard method.
(SD years) is incorrect. SD has not unit.

Please, remove the second hand REFS. such as ghasemi...
please,cite tge primary Refs.

Validity of the findings

The method and result section need organization.

Additional comments

Discussion must be related to the results.
However, there are several results without discussed.
The limitation must be revised.

Reviewer 3 ·

Basic reporting

Clearer info described the workflow of taught and process. The literature references are provided adequately. Self-contained with relevant results to the hypothesis.

Experimental design

Methods described with sufficient detail and information to replicate.

Validity of the findings

All the underlying info has been provided. They are robust, statistically sound, and well presented.

Additional comments

In principle, I satisfy with all additional info and correction done. The info is clearer and had back up with the works of literature. I have no doubt to accept this piece of work for publication in this prestigious journal.

---

## Round 0.3 · Minor Revisions

Thank you for the resubmission. Please address the concerns of the reviewers and ensure that your article is thoroughly proof read before your next submission.

Reviewer 1 ·

Basic reporting

Line 117, add " there is" to no standard and validated ....
Line 128, change the word "received" to "provided". The participants received informed consent...
Line 138: a university student, and have started smoking regularly for at least one year. Please be more specific, the word “regularly” is quite vague a better description would be for example smoke daily or at least 1 pack of cigarettes a day would be more specific
Line 181-182: QSU 12-items Persian version was finalized after some minor revisions and corrections... so what are the revisions/correction give a few examples.
Line 251-251: The CFI value in the single-factor model was not good (.76), it was very good in the modified two-factor model (.97), and it was not acceptable in the original two-factor model (.86). These sentence could be rearranged in a better understandable manner.
For example, the CFI values for single-factor (.76), modified two-factor model (.97) and original two-factor model (.86) respectively. The general acceptable CFI value should be above .95 which is achieved by the modified two-factor model in our CFA analysis.

Figure 1, the legend is missing to explain what the numbers represent.

Line 273-274: The Cronbach’s alphas for Factor 1 (α = .94, mean scores = 5.55±.13) and Factor 2 (α = .97, mean score = 5.31±.48), and for the total construct (α = .95, mean scores = 5.41±.55)
Mean and SD are not needed here…

Line 292: Do you mean concurrent validity as opposed to convergent validity?
“are, the higher the nicotine dependency is. That is, indicating a fair convergent validity between…”

Line 388: negativly associate- missing an “e”
Line 404: Nevertheles,- missing an “e”
Line 447: Psyciatric- typo

Experimental design

No comment.

Validity of the findings

Conclusion are well stated but a few redundant words in line 566-567. Line 571-573. Kindly double check.

I would like to suggest another round of proof reading as there are grammatical and structural throughout this manuscript.

Reviewer 2 ·

Basic reporting

Currently, the manuscript has significantly promoted during review rounds.

Experimental design

I suggest the new title:
Psychometric Validation Currently, the manuscript has significantly promoted during review rounds

Validity of the findings

Currently, the manuscript has significantly promoted during review rounds

Additional comments

Currently, the manuscript has significantly promoted during review rounds

---

## Round 0.4 · accepted · Accept

Thank you for your revised version of the paper, which I am pleased to recommend for publication. Congratulations!